# Comparison of Optogalvanic and Laser-Induced Fluorescence Spectroscopy

**Laurentius Windholz**

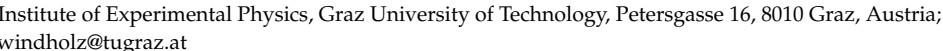

Institute of Experimental Physics, Graz University of Technology, Petersgasse 16, 8010 Graz, Austria; windholz@tugraz.at

**Abstract:** When investigating complex atomic spectra, it may happen accidentally that two or even several transitions between different pairs of combining energy levels have nearly the same wavenumber, and the observed spectral lines are overlapping (blend situations). In such cases, investigations of hyperfine structures can be very helpful in the identification of the involved transitions. In this paper, two complicated blend situations within the spectra of lanthanide atoms (Praseodymium and Lanthanum) are discussed as examples. The experimental methods applied are optogalvanic and laser-induced fluorescence spectroscopy, combined with emission spectra gained via Fourier transform spectroscopy. It is shown that, in such cases, a combination of optogalvanic and laser-induced fluorescence detection is necessary to find all transitions contributing to the observed spectral signatures.

**Keywords:** laser spectroscopy; optogalvanic spectroscopy; laser-induced fluorescence; Praseodymium; Lanthanum





## 1. Introduction

The investigations of our group in Graz were directed, on one hand, to the determination of hyperfine (hf) structure constants of several chemical elements and, on the other hand, to the discovery of previously unknown energy levels.

In general, the emission spectra of atoms provide insight into the structure of their electronic shell. But even using the most recent computing techniques, it is not possible to describe the electron shell with the help of ab initio methods. Thus, for the prediction of still-unknown energy levels and their data (e.g., Landé $g_j$—factors and hyperfine (hf) structure constants), semi-empirical methods must be applied [1,2]. These methods are based on fitting the radial part of the wave function to experimentally known data. Thus, their accuracy is strongly dependent on reliable data, which should be as comprehensive as possible.

Another motivation for investigating the spectra of complex atoms and their hf structure is the fast development of space-based high-resolving spectrometers, which need verified atomic data for interpretation of the observations (see, e.g., [3]). For example, spectral lines of Lanthanide atoms are observed in cooler chemically peculiar (CP) stars [4].

Within the spectra of complex atoms, quite often, overlapping spectral lines are observed, so-called blend situations. In such cases, the energy differences of different pairs of combining levels have accidentally nearly the same value, and the observed spectral lines are very close to each other or even overlapping. If such lines have widely split hf structures, these spectral signatures can be used to identify the transitions contributing to the observed complicated patterns.

Complicated blend situations are quite often observed in the highly resolved spectra of lanthanide atoms, due to their huge number of energy levels. Thus, here, the combination of optogalvanic (OG) and laser-induced fluorescence (LIF) detection is discussed using examples within the spectra of atomic Praseodymium (Pr I) and atomic Lanthanum (La

I). It is shown how to find all blend lines and how to determine which energy levels are involved in the formation of the lines. Through a combination with Fourier transform (FT) spectroscopy, accurate wavelengths can be obtained, even for weak lines which are not noticeable in the FT spectrum.

Apart from blend situations, OG spectroscopy can significantly enhance sensitivity compared to emission spectroscopy. This is especially the case when high-lying energy levels are investigated, which usually do not contribute much to spectral emission, as was shown, e.g., in Figure 2 of ref. [5].

For atomic Pr (Pr I), about 1100 even-parity levels and 1500 odd-parity levels are currently known. The databank contains more than 9000 Pr I—lines. The manifold of La I is a little bit smaller; now, about 550 even-parity La I—levels and 220 odd-parity levels are known, and in the databank, approximately 6000 La I—lines are contained. For the handling of these data, the program "Elements" [5,6] is used, which provides certain wavelength classification suggestions and predicts hf patterns, if the hf constants of the combining levels are known. Nevertheless, for all investigated chemical elements, there also exist a huge number of spectral lines which, up to now, cannot be classified; thus, by far, not all energy levels are known.

In the present article, it is shown that OG and LIF detection must be both used to identify all lines which contribute to a complicated blend situation.

## 2. Experiment

Hollow cathode lamps were and are widely used as light sources in classical and FT spectroscopy. For performing laser spectroscopy, a special home-made see-through hollow cathode lamp is used, which is based on the ideas of H. Schüler [7] and was further modified by the group of Prof. Dr. Guthöhrlein in Hamburg [8,9]. This type of hollow cathode lamp was used by the group in Graz and is further used by groups at the University of Technology Poznan (Poland), University of Gdansk (Poland), and University of Istanbul (Turkey).

The lamp consists of a grounded cathode, ca. 20 mm long, with an inner diameter of 3 to 4 mm, and two anodes, with an axial distance of 0.8 to 1.2 mm, on both sides of the cathode (see Figure 1a; ceramic holders and voltage pins are not shown). The cathode is made either from the material to be investigated (e.g., Ta [10]) or from copper. In the latter case, the bore is bushed with a thin layer of the metal of interest (e.g., Pr or La) or is prepared with a powder containing a stable chemical compound of it (e.g., $Ta_2O_5$ [11]). The discharge region is cooled with the help of liquid nitrogen in order to minimize Doppler broadening of the spectral lines. The discharge is usually operated in a constant current mode (20 to 90 mA), and the voltage is ca. 350 V. First, the hollow cathode lamp is evacuated to ca $10^{-3}$ mbar, and then it is filled with a noble gas (Ar or Ne, ca. 0.2 to 1.5 mbar) and an electric voltage is applied. After ignition, the plasma burns in the noble gas, and the lamp emits a weak light that is characteristic of Ar or Ne. After some minutes, a sputtering process begins, and the discharge is mainly carried by metal atoms, which have a much lower ionization limit than the noble gases. Finally, the discharge becomes stable, is burning mainly in the metal vapor, and now emits a bright light that is characteristic of the element under investigation (e.g., green for Pr and white for La).

A sketch of the laser spectroscopic apparatus is shown in Figure 1b. A narrow-band tunable laser light is provided by a suitable laser system. Here, a home-made cw dye laser, based on the optical scheme of a Coherent 699-21 laser, but with a scan range slightly more than 45 GHz, was used. The laser wavelength was measured with a home-made lambdameter, with a resolution and accuracy of 0.01 Å [12]. A small part of the laser light was provided to a home-made, temperature-stabilized confocal Fabry–Perot interferometer (marker etalon). Its transmission signal (free spectral range 367.33(6) MHz) was used to convert the records, made by scanning the laser frequency, from signal versus time to signal versus offset frequency. The main part of the laser light, modulated by a mechanical chopper, intersected the hollow cathode. The discharge light, emitted in the direction

opposite to the incoming laser beam, was captured by a mirror with a hole and a lens and focused to the entrance slit of a monochromator. At the exit slit of the monochromator, a photomultiplier detected the transmitted light. A lock-in amplifier, synchronized with the chopper, was used to amplify the changes in the transmitted light due to action of the laser beam with the plasma (laser-induced fluorescence, LIF). The lock-in amplifier could be also used to amplify changes in the voltage drop on a resistor in the discharge circuit, in order to record so-called optogalvanic (OG) signals. One can quickly switch between OG and LIF signals, even during a laser scan. With this arrangement, Doppler-limited spectra can be recorded. By changing the optical path of the exciting laser light, Doppler-reduced spectra (via saturation spectroscopy) can also be obtained.

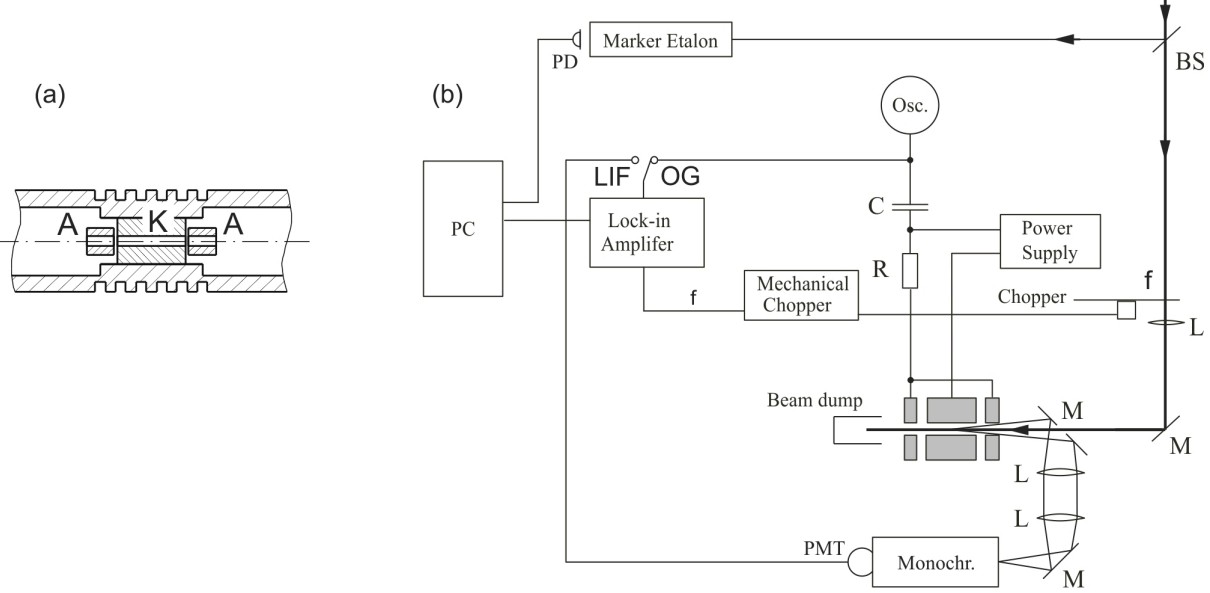

**Figure 1.** Experiment. (**a**) Schematic cut through the hollow cathode. A—anodes, K—cathode (grounded). (**b**) Simplified sketch of the apparatus. BS—beam splitter, M—mirror, L—lens, PMT—photomultiplier tube, PD—photodiode, R—resistor, C—capacitor, Osc.—oscilloscope, f—chopper frequency, PC—personal computer.

The discharge plasma is a self-luminous source of free atoms, and the population of the energy levels follows a detailed thermodynamic equilibrium. This has the advantage that levels having relatively high energy are also sufficiently populated and can serve as lower levels of laser-driven transitions. On the other hand, usually, laser excitation does not influence the equilibrium population very much; thus, the additional LIF intensity caused by laser excitation may be small compared to the plasma emission intensity. During OG detection, laser excitation causes only very small changes in the discharge voltage (some mV versus hundreds of V). Thus, usually, small changes in a high signal must be detected, which makes the use of phase-sensitive detection unavoidable. This high background signal causes additional noise.

### 3. A Blend Situation within the Pr I—Spectrum

Praseodymium, atomic number 59, has one stable isotope with mass number 141 and nuclear spin quantum number $I = 5/2$. Its magnetic moment is relatively large (4.2754(5) $\mu_N$ [13]), but its electric quadrupole moment is small ($-0.077(6)$ b [13]). Thus, the hyperfine (hf) structure of Pr lines is dominated by the magnetic hf structure constant $A$, and only for some energy levels values of the electric hf structure constant $B$ can be determined from Doppler-limited recordings.

The ground level of Pr has odd parity, and higher odd-parity levels start at energy 1376 cm$^{-1}$. The lowest even-parity level has an energy of only 2793 cm$^{-1}$; thus, the ladders

of even and odd levels run nearly parallel. This fact and the existence of a huge number of even and odd levels causes a very rich spectrum of Pr; at nearly each laser frequency, more than one transition is excited, since some differences in energy between pairs of combining levels usually coincide. All excited transitions contribute to the OG signal, and the relative strengths of the signals are dependent on their influence on the discharge current. This influence is—among others—dependent on the transition probability between the combining levels, the population of the lower level of the excited transition, and the position of the upper level relative to the ionization limit.

On the other hand, LIF detection is selective to the excited levels. But, when investigating a certain region of the Pr spectrum, it is difficult to find all possible LIF lines. This must be performed for each laser wavelength by scanning the monochromator transmission wavelength. Thus, one can benefit from a highly resolved FT spectrum or an OG spectrum for setting appropriate excitation wavelengths and then searching for LIF signals.

A small part of the Pr spectrum, between 5838.04 and 5837.47 Å (in the whole paper, air wavelengths are used), is discussed now as an example of the combination of FT, OG, and LIF spectroscopy.

In Figure 2, trace (a), the OG spectrum is shown in dependence on the offset frequency. The start wavelength of the laser scan is 5838.04 Å (offset frequency 0), and the scan direction is to higher laser frequencies (and thus lower wavelengths).

In the investigated region, one strong spectral line is known, classified already by A. Ginibre [14,15]. This line is clearly visible in an FT spectrum [16], trace (c), with a signal-to-noise-ratio (SNR) of 140. A simulation of the hf pattern of this line is shown in trace (b), directly within trace (a), using the known hf constants of the involved levels.

A. Ginibre also noticed the peak at ca. 27,500 MHz as a separate line but could not classify it. On the left side of the strong line, another line with an SNR of 10 can be identified. The hf pattern of this line is shown in Figure 2 as trace (h). But the peak at ca. 27,500 MHz offset frequency is too high in the FT spectrum; thus, another line must be involved. Through the use of the classification program "Elements", this peak was identified and recorded as trace (g). The lower level of line (h) is higher than that of line (g) and thus a little bit less populated. Nevertheless, the LIF signal of line (g), recorded at an LIF wavelength of 4934 Å (SNR in the FT spectrum 12), has a relatively low SNR of 15, while the LIF signal of line (h), recorded at an LIF wavelength of 5468 Å (SNR in the FT spectrum 10), has a much higher SNR of 50. A possible reason may be the different branching ratios of the upper, laser-populated levels of lines (g) and (h). Nevertheless, not all structures visible in the OG spectrum at offset frequencies <20,000 MHz are explained by lines (g) and (h). Apparently, some still-undetected lines contribute to the OG signal.

Comparing traces (a) and (b) in Figure 2, it is clear that, at the high-frequency part of the strong line, some other lines must contribute to the OG signal of trace (a). Outside the laser spectroscopy-investigated range, line (d) is present, having two weak components in this range. Two further lines were found by laser excitation and the detection of LIF, shown in Figure 2 as (e) and (f). The lower and upper levels of both lines are different only by ca. 10 cm$^{-1}$. Despite this fact, the SNR of both detected LIF lines differs by a factor of ca. 2. But at offset frequencies > 40,000 MHz, lines (e) and (f) do not sufficiently explain the OG signal. The used classification program does not suggest a line with suitable cg wavelength and hf structure, which can explain the remaining deviation. Thus, most probably, a transition to a yet-unknown Pr level is excited when recording the OG signal.

The classification of all Pr lines shown in Figure 2, including the hf constants of the combining levels, is given in Table 1. A level scheme showing the lines discussed is shown in Figure 3. For lines (b) and (d), the wavelength could be determined from the wavenumber-calibrated FT spectrum with high accuracy (uncertainties ±0.0001 Å), while for lines (g) and (h), the uncertainties are larger (±0.001 Å). For lines (e) and (f), the wavelength uncertainties are ±0.01 Å due to the properties of the used lambdameter.

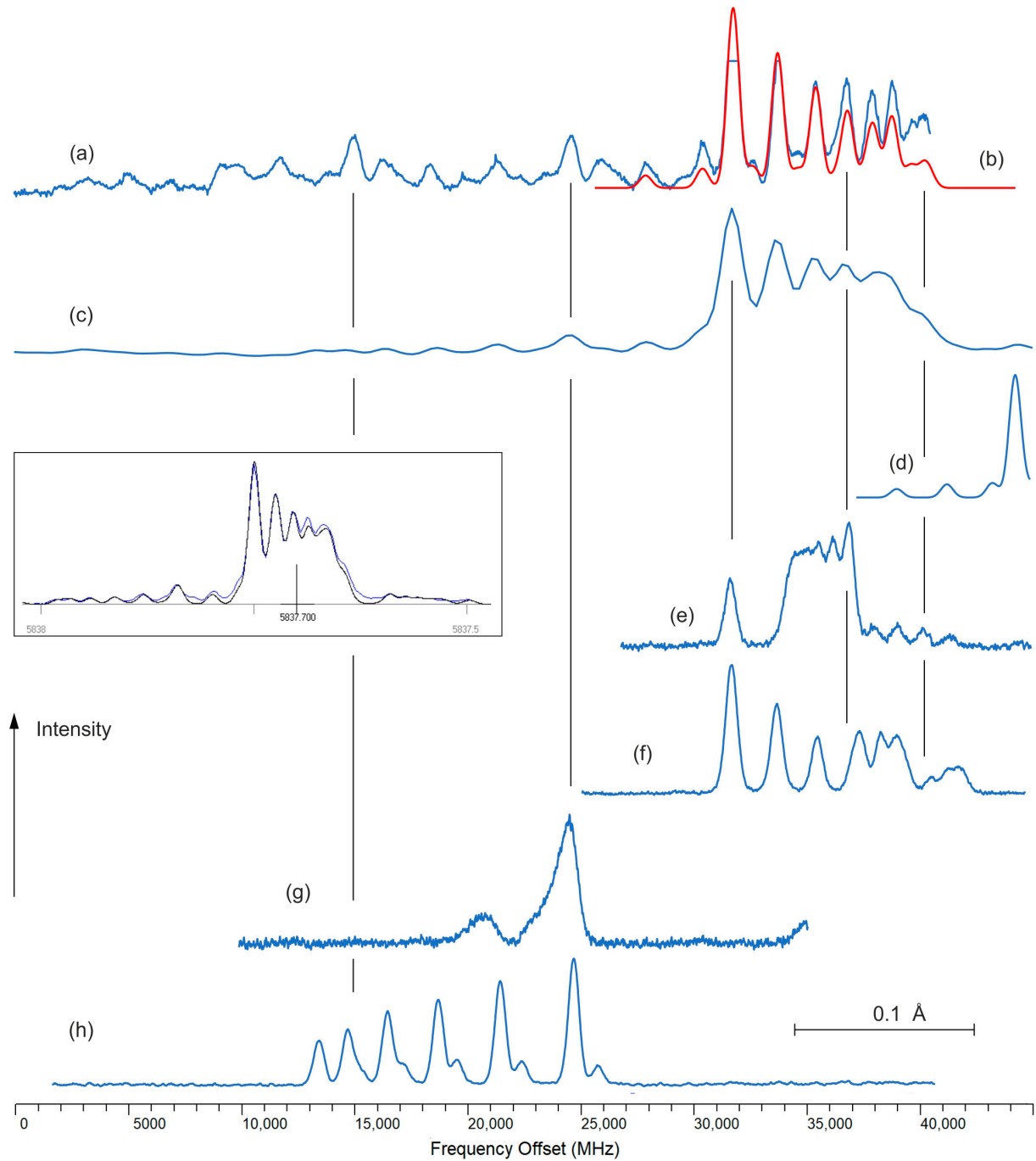

**Figure 2.** Optogalvanic, FT, and LIF spectra between 5838.04 and 5837.47 Å. Classification of lines: see Table 1. FWHM—full width at half maximun, cg—center of gravity, wl—wavelength. The vertical lines should guide the eye of the reader. The inset shows the FT spectrum and a simulated spectrum (black) taking into account lines (d,e,g,h). FWHM = 1300 MHz, close to the resolution of the FT spectrum. (a) Optogalvanic record. (b) Simulation of the strong line at cg wl 5837.7101 Å using hf constants known from literature assuming FWHM = 700 MHz. (c) FT spectrum. FWHM of the observed structures 1300 MHz. Here, only the strong lines (b) and line (h) are visible. (d) This line (cg wl 5837.5607 Å) has some weak hf components in the investigated region. Shown is a simulation assuming FWHM = 700 MHz. (e,f) Hidden under the strong line at 5837.7101 Å two further transitions were found, detected via LIF. (g,h) The peak at ca. 27,500 MHz offset frequency, visible in the FT spectrum and the OG spectrum, is caused by the overlap of two further lines, detected via LIF.

**Table 1.** Pr-transitions shown in Figures 2 and 3. SNR FT—signal-to-noise-ratio in the FT spectrum, wl—wavelength, *E*—energy, *J*—angular momentum, *A*,*B*—hf structure constants, Ref—reference to values of *A*, *B*.

| Figure 2 Trace | wl (Å) | SNR FT | Odd E (cm⁻¹) | J | A (MHz) | B (MHz) | Ref. | Even E (cm⁻¹) | J | A (MHz) | B (MHz) | Ref. | LIF Detected at wl (Å) |
|---|---|---|---|---|---|---|---|---|---|---|---|---|---|
| (b) | 5837.7101 | 140 | 0.000 | 9/2 | 926.209 | −11.878 | [17] | 17,125.256 | 9/2 | 614.0(3) | −4(4) | [18] | |
| (d) | 5837.5610 | 10 | 28,675.293 | 9/2 | 831(5) | | [19] | 11,549.602 | 9/2 | 1064(2) | | [20] | |
| (e) | 5837.67 | - | 12,108.867 | 5/2 | 1272(2) | | [21] | 29,234.227 | 3/2 | 1038(4) | | [22] | 5525.571 |
| (f) | 5837.70 | 8 | 29,243.323 | 13/2 | 638(5) | | [23] | 12,118.039 | 13/2 | 554(1) | −45(30) | [20] | 5268.121 |
| (g) | 5837.850 | 9 | 29,742.537(20) | 9/2 | 777(8) | | [24] | 12,617.700 | 7/2 | 883(2) | | [25] | 4934.694 |
| (h) | 5837.903 | 10 | 30,485.194(20) | 9/2 | 684(3) | | [24] | 13,360.511 | 11/2 | 151(3) | | [26] | 5468.668 |

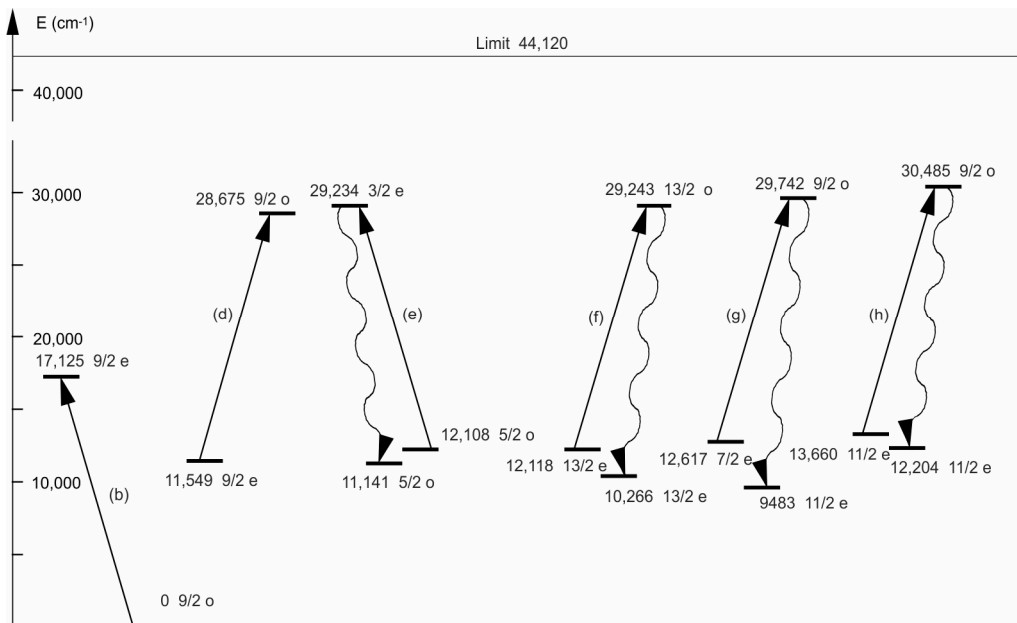

**Figure 3.** Level scheme of Pr I corresponding to Figure 2. The transitions are labelled according to Table 1. (b) Strong line dominating the FT spectrum; (d) line at the high-frequency border of Figure 2; (e–h) lines recorded with LIF. The LIF signals have the same phase as the OG signal; the upper level of the LIF lines is populated via laser light.

## 4. A Blend Situation within the La I—Spectrum

Lanthanum has an atomic number of 57, and its natural composition is dominated at the amount of 99.91% by a stable isotope, with mass number 139 and nuclear spin quantum number $I = 7/2$. Like in Pr, the magnetic moment is relatively large (+2.7830455(9) $\mu_N$ [13]), but its electric quadrupole moment is small (0.200(6) b [13]). The hyperfine (hf) structure of La lines is again dominated by the magnetic hf structure constant *A*.

Also, La I has a huge number of energy levels. The ground level has even parity, and the higher even-parity levels start at an energy of 1053 cm⁻¹. But, here, the lowest odd-parity level has an energy of 13,260 cm⁻¹. Thus, transitions from even to odd and from odd to even are not as strongly mixed as in Pr, and the number of lines is smaller. Nevertheless, a huge number of separated lines for this element's complicated blend situations are also observed, and one of them is discussed here.

In Figure 4, the spectral region between 5649.48 Å (offset frequency 0) and 5649.08 Å (offset frequency of 50 GHz) is treated. In trace (a), the FT spectrum [27] is shown. Only one peak, at ca. 12,000 MHz offset frequency, is noticeably larger than noise (SNR ≈ 2), but this peak allows one to find an accurate cg wavelength value (5649.3797 Å) for the corresponding spectral line.

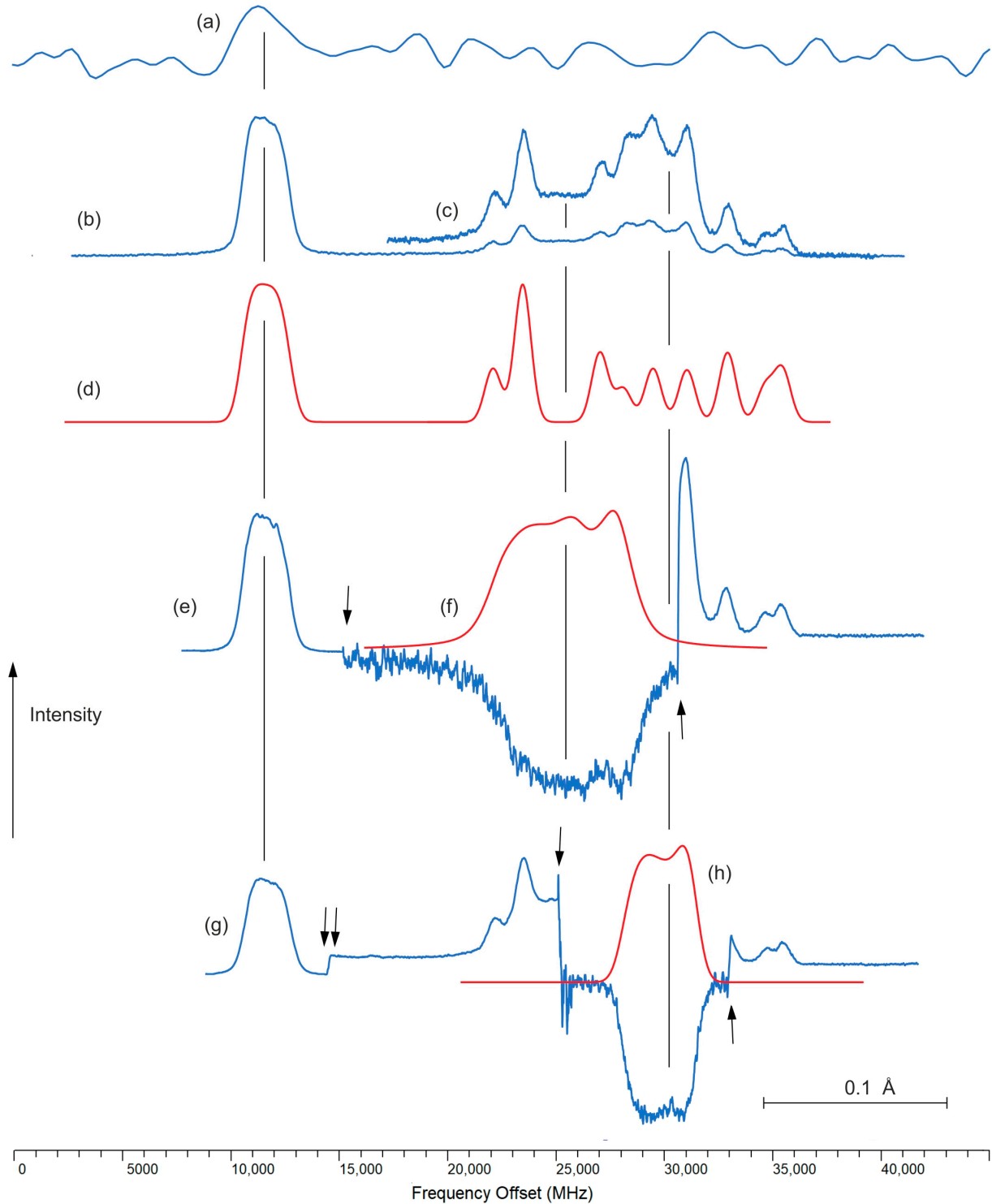

**Figure 4.** Optogalvanic, FT, and LIF spectra of La between 5649.48 and 5649.08 Å. Classification of lines: see Table 2. SNR—signal-to-noise-ratio, FWHM—full width at half maximum, cg—center of gravity, wl—wavelength. (a) FT spectrum. Only the strong line at 5649.3797 Å is visible with SNR ≈ 2. (b) OG spectrum. The strong line is shown here with SNR ≈ 80. Additionally, a complicated blend structure is visible at higher frequencies. (c) The blend structure is shown with the same height as the strong peak. SNR ≈ 25. From the classification program, it is easy to find out that the outermost structures of the blend belong to a wide split $J_{up} = 3/2$ to $J_{low} = 3/2$ transition. (d) Simulation of the strong line at cg wl 5649.3797 Å and the $J_{up} = 3/2$ to $J_{low} = 3/2$ transition (cg wl 5649.184 Å) using hf

constants known from the literature, assuming FWHM = 1000 MHz. Both structures are normalized to the same intensity. (e) For use as wavenumber marker, first, the OG signal of the strong line was recorded. At $\downarrow$, it was switched to the LIF signal (monochromator wavelength 4648 Å), and after recording the line, at $\uparrow$, back to OG recording (with higher amplification). (f) Simulation of the LIF line, using a line profile which is a sum of 80% Gaussian and 20% Lorentzian shape, both with FWHM = 2000 MHz. (g) Again, first, the strong line was recorded using OG, and at $\downarrow\downarrow$, it was switched to higher amplification. At $\downarrow$, it was switched to the detection of the LIF signal (monochromator wavelength 5145 Å), and after recording the line, at $\uparrow$, back to OG recording. (h) Simulation of the LIF line, using a Gaussian line with FWHM = 1200 MHz. As can be seen, the LIF signals have opposite sign compared to the OG signals. This is an indication that the population of the upper level of the LIF line is lowered by laser excitation. This level is, at the same time, the lower level of the excited transition.

Trace (b) shows the same region recorded with the help of the OG signal. The line at 5649.3797 Å shows up now with SNR = 80, and at its high-frequency side, a complicated blend situation appears with SNR = 25. This part is enlarged and shown as trace (c). The strong line has a very narrow hf structure with overlapping components (transition from $J_{low} = 9/2$ to $J_{up} = 9/2$), while the main structure of the blend could be easily classified as a transition from $J_{low} = 3/2$ to $J_{up} = 3/2$ with widely split, well-separated components. A simulation of both lines (the intensities normalized to the same height, FWHM = 1100 MHz) is shown in trace (d).

Then, the laser wavelength was set to regions in which the OG signal of the blend was not explained by the 3/2–3/2 transition, and a search for LIF signals was performed by scanning the monochromator transmission wavelength. No LIF signals with the same phase as the OG signal were observed. Apparently, the upper laser-excited levels are depopulated mainly by collisions, which lead to ionization. Thus, a relatively strong OG signal, but no LIF-signal, is observed. Such behavior is observed for most Li I—levels with energies higher than 40,000 cm$^{-1}$.

But, at some wavelengths, LIF-signals with the opposite phase occurred. Such a "negative" LIF line (nLIF) occurs if the laser light depletes the population of the upper level of the observed LIF line; this is, at the same time, the lower level of the laser-driven transition. During all investigations, it turned out that nLIF-lines are strong, well-known La transitions to low-lying even-parity levels. Thus, the observation of an nLIF-line reveals the upper, odd-parity level of the observed nLIF-line. In this way, the lower level of the laser-excited transition is identified, and the unknown upper level must have even parity. The *J*-value and the hf-constants of such a level then can be determined from the hf pattern of the excited line.

In trace (e), one of the two observed nLIF signals, at wavelength 4648 Å, is shown. First, the OG signal was recorded; at offset position $\downarrow$, the detection was switched to LIF detection, and at $\uparrow$, it was switched back to OG (with higher amplification). This allows us not only to record the hf pattern of the unknown blend line but also to determine the exact frequency position relative to line (b). Trace (f) shows a simulation using the known hf constants and a line profile which is a sum of 80% Gaussian and 20% Lorentzian shape, both with FWHM = 2000 MHz.

Trace (g) shows the third line contributing to the recorded OG signal, recorded at an nLIF-wavelength of 5145 Å. At $\downarrow\downarrow$, the lock-in amplifier was switched to higher amplification; at $\downarrow$, to the detection of LIF signals; and, at $\uparrow$, back to OG. The line has a small hf pattern and lies completely within the 3/2–3/2–transition. A simulation is shown in trace (h), using a Gaussian profile with FWHM = 1200 MHz.

As can be learned from Figure 4, for simulating the three lines of the blended one requires different line profiles. This can be explained by different broadening mechanisms occurring in the upper levels, as pointed out in ref. [28]. The SNR of the nLIF-lines is not very high (8 and 12, respectively). This is explained by the fact that the upper level of the strong nLIF-line is highly populated by the discharge, while the laser light depopulates the level only slightly.

A level scheme illustrating the description of Figure 4 is shown in Figure 5. All lines discussed are classified in Table 2.

As can be seen from Figure 4, trace (a), the wavenumber-calibrated FT spectrum allows to find the cg wavelength of the line at 5649.3797 Å with high precision (uncertainty 0.0001 Å). On the other hand, the laser wavelength could be measured only with an accuracy of 0.01 Å (see Section 2). But, when fitting the observed spectra shown in Figure 4, traces (b), (e), and (g), the offset between line (b), and the three blend lines could be determined, leading to the wavelength values given in Table 2 (uncertainties 0.001 Å).

**Table 2.** La-transitions shown in Figures 4 and 5. SNR FT—signal-to-noise-ratio in the FT spectrum, wl—wavelength, *E*—energy, *J*—angular momentum, *A*, *B*—hf structure constants, Ref—reference to values of *A* and *B*.

| Figure 4 Trace | wl (Å) | SNR FT | Odd $E$ (cm$^{-1}$) | $J$ | $A$ (MHz) | $B$ (MHz) | Ref. | Even $E$ (cm$^{-1}$) | $J$ | $A$ (MHz) | $B$ (MHz) | Ref. | nLIF Detected at wl (Å) |
|---|---|---|---|---|---|---|---|---|---|---|---|---|---|
| (b) | 5649.3797 | 2 | 21,383.990 | 9/2 | 33.9(30) | | [29] | 39,080.131 | 9/2 | 94.9(10) | −20(15) | [30] | |
| (c) | 5649.184 | - | 17,797.298 | 3/2 | 1335.0(10) | | [31] | 35,494.057 | 3/2 | 301(1) | 25(7) | [32] | |
| (f) | 5649.213 | - | 24,173.826 | 3/2 | −228.9(22) | 30(11) | [33] | 41,870.494 | 5/2 | 165(5) | | [34] | 4648.639 |
| (h) | 5649.168 | - | 22,439.346 | 3/2 | 149.5(32) | −45(35) | [33] | 40,136.158 | 5/2 | 243.8(10) | | [31] | 5145.418 |

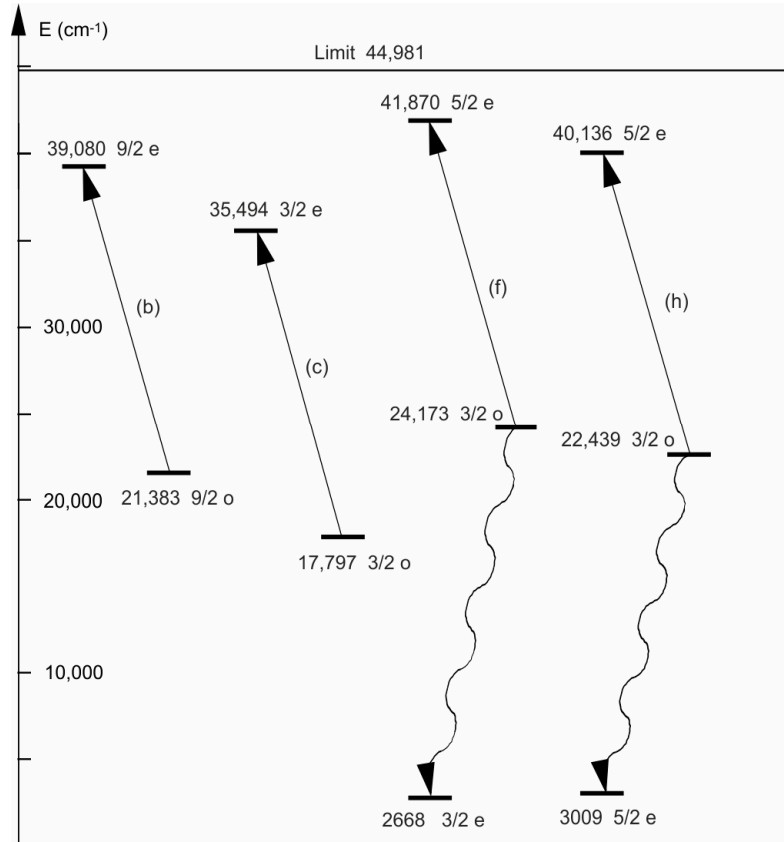

**Figure 5.** Level scheme of La I corresponding to Figure 4. The transitions are labelled according to Table 2. (f,h) Lines recorded with LIF. The LIF signals have the opposite phase as the OG signal; the upper level of the LIF lines is de-populated via laser light, and their intensity is lowered. La I—levels higher than 40,000 cm$^{-1}$ are most probably ionized in La-plasma, and no direct decay lines of such levels can be observed.

## 5. Discussion

The main points of how the lines forming the blends described in Sections 3 and 4 were found are discussed in the respective sections. If a line is visible in the FT spectrum,

the laser wavelength can be set to this value, and a search for LIF lines, in order to identify the transition, leading to the respective peak in the FT spectrum, can be carried out. This method was used mainly when investigating the spectrum of Pr I, for which, in some regions, the FT spectrum is so dense that practically no space between overlapping hf patterns is left. A huge number of previously unknown energy levels, having both even and odd parity, was discovered, partly published in the references on Pr given here [16,20–22,24,25], together with their hf constants.

For La I, the lines visible in the FT spectrum are usually well separated and, in most cases, do not show blend situations. But, when performing a continuous OG scan over a wider spectral range (overlapping laser scans), a huge number of additional transitions to high-lying even-parity levels were detected, leading, in most cases, to the discovery of yet-unknown energy levels. An example of this is given in ref. [5]. Previously unknown energy levels, together with their hf constants, are partly published in the references [5,27,28,31,32,34].

## 6. Conclusions

It was shown that a general decision between the application of OG or LIF spectroscopy cannot be made. In complicated blend situations, both methods have to be applied. FT spectra and OG spectra are well suited to set the laser wavelength to a value, where unexplained spectral features are observed. Finding all lines contributing to a blend situation needs level-sensitive methods like recording LIF signals.

If spectra recorded via LIF or OG show only one well-separated line, in some cases, the determination of the hf constants of both combining levels is possible, and at least one of the involved levels can be identified by its hf constants. An overview of how unknown energy levels can be found is given in ref. [5].

For the majority of investigated lines, and especially for blend situations, it is necessary to identify at least one of the levels involved in a transition by treating the manifold of LIF lines. LIF signals that have a positive phase (intensity increased by laser interaction) mark the upper level of the laser-excited transition, as discussed using the example of the Pr-blend. If the intensity of an LIF line is lowered by laser interaction, the lower level of the driven transition is marked, as in the discussed La-blend.

Usually, a combination of all observations is necessary to determine which transitions are observed, together with the introduction of previously unknown energy levels.

**Funding:** This research received no external funding. Open Access funding is provided by Graz University of Technology.

**Institutional Review Board Statement:** Not applicable.

**Informed Consent Statement:** Not applicable.

**Data Availability Statement:** Data available from author (windholz@tugraz.at).

**Acknowledgments:** I would like to thank all members of my former group for contributions to the hf spectroscopy of Pr and La, especially Shamim Khan, Bettina Gamper, Imran Siddiqui, and Tobias Binder.

**Conflicts of Interest:** The author declares no conflict of interest.

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
