# Peer review of "Comparison of Optogalvanic and Laser-Induced Fluorescence Spectroscopy"

_photonics, doi:10.3390/photonics11030279_

Round 1

Reviewer 1 Report

Comments and Suggestions for Authors

The paper explains the need for both Opto-galvanic and Laser-Induced Spectroscopy, especially in blend lanthanide systems. The two methods are known to play complementary roles in unraveling intricate hyperfine spectral features. This manuscript in its current condition lacks a general overview of the importance and implications of the data. A few points should be addressed in the revised manuscript for a broad readership.

(i)                       Practical and Contemporary interest in the general field, and what are the implications in other systems? Not only in Lanthanides, the discrepancy of a single technique, especially the limitations of LIF has been discussed even in noble gas systems such as neon. Glow discharge is used in various applications such as Plasma processing. A more general and comprehensive introduction to these observations is needed for a broad readership.

(ii)                    Except for the blend situations, how does the combination of these techniques yield precise information compared to the individual ones?

(iii)                  The author should explain in more detail how the spectroscopic diagnostics of the plasma reveals new knowledge in Lanthanide systems.

Comments on the Quality of English Language

Figure Captions can be improved.

Author Response

Thank you for the valuable comments and suggestions.

Response see attached letter.

Reviewer 2 Report

Comments and Suggestions for Authors

Comments:

1.     Line 6, “within Pr I and La I it is shown that in” à “within Praseodymium (Pr) I and Lanthanum (La) I it is shown that in”,  for first appearance in the text.

2.     Line 56, “based on the optical scheme od a Coherent 699-21 laser” à “based on the optical scheme of a Coherent 699-21 laser”

3.     Line 59, “(free spectral range 367.33(6) MHz)” please add a reference.

4.     Line 96, “1376 cm-1. The lowest even-parity has an energy of only 2793 cm-1, thus the ladders of” à “1376 cm-1. The lowest even-parity has an energy of only 2793 cm-1, thus the ladders of”. Superscript for “cm-1” and in all texts.

5.     Line 110, “A smal part of the Pr spectrum, between 5838.04 and 5837.47 Å, is discussed now as”, one should indicate the wavelength, 5838.04 and 5837.47 Å, is in air or in vacuum? And “smal” should be “small”.

6.     Lines 141-143, Caption of Fig. 2, “Fig. 2 Optogalvanic, FT and LIF spectra between 5838.04 and 5837.47 Å. Classification of lines: see Table 1. FWHM … full width at half maximun, cg … center of gravity, wl … wavelength. The vertical lines should guide the eye of the reader. The inset shows the FT spectrum and a simulated spektrum (black) taking into account lines d), e), g), and h). FWHM = 1300 MHz close to the resolution of the FT spektrum.” à “Fig. 2 Optogalvanic, FT and LIF spectra between 5838.04 and 5837.47 Å. Classification of lines: see Table 1. The inset shows the FT spectrum and a simulated spectrum (black) taking into account lines d), e), g), and h). FWHM = 1300 MHz close to the resolution of the FT spectrum.”. please change the spektrum to spectrum for consistency.

7.     Line 222, “full width at half maximun” à  “full width at half maximum”.

Author Response

(The authors gave the same response as above.)

Reviewer 3 Report

Comments and Suggestions for Authors

The manuscript describes a hfs method to detect overlapping spectral lines when blends occur in spectra of heavy elements with relatively high level density, using two examples of Pr I and La I. I found the work very informative and of great importance to the spectroscopy community. The manuscript is very well written and the conclusions are sufficiently supported by the results to warrant publication. Nevertheless, I have three minor points that I would ask the author to address in the final version:

- For a better understanding of the current state of research and the necessity of the method used, I recommend adding one to two references in the introductory section

- Could the author check proper formatting of "cm-1", "Jup", "µN" etc.

- Could the author check the typos in: Line46: "hollow cathode"; Line56: "of a Coherent"; Line90: "and nuclear spin"; Line154: "g) and h)" instead of "e) and f)"; Line240: "at offset position xxx where"

Author Response

(The authors gave the same response as above.)

Round 2

Reviewer 2 Report

Comments and Suggestions for Authors

The revised manuscript improves their presentation of the scientific problem and is ready for publication in the journal Photonics.